# Layer-by-layer phase transformation in $Ti_3O_5$ revealed by machine-learning molecular dynamics simulations

Mingfeng Liu[1,2,6], Jiantao Wang[1,2,6], Junwei Hu[3,6], Peitao Liu[1] ✉, Haiyang Niu[3] ✉, Xuexi Yan[1], Jiangxu Li[1], Haile Yan[4], Bo Yang[4], Yan Sun[1], Chunlin Chen[1], Georg Kresse[5], Liang Zuo[4] & Xing-Qiu Chen[1]

Reconstructive phase transitions involving breaking and reconstruction of primary chemical bonds are ubiquitous and important for many technological applications. In contrast to displacive phase transitions, the dynamics of reconstructive phase transitions are usually slow due to the large energy barrier. Nevertheless, the reconstructive phase transformation from $\beta$- to $\lambda$-$Ti_3O_5$ exhibits an ultrafast and reversible behavior. Despite extensive studies, the underlying microscopic mechanism remains unclear. Here, we discover a kinetically favorable in-plane nucleated layer-by-layer transformation mechanism through metadynamics and large-scale molecular dynamics simulations. This is enabled by developing an efficient machine learning potential with near first-principles accuracy through an on-the-fly active learning method and an advanced sampling technique. Our results reveal that the $\beta-\lambda$ phase transformation initiates with the formation of two-dimensional nuclei in the $ab$-plane and then proceeds layer-by-layer through a multistep barrier-lowering kinetic process via intermediate metastable phases. Our work not only provides important insight into the ultrafast and reversible nature of the $\beta-\lambda$ transition, but also presents useful strategies and methods for tackling other complex structural phase transitions.

Solid–solid phase transitions are arguably one of the most ubiquitous phenomena in nature and have important implications for numerous scientific areas such as metallurgy[1], ceramics[2,3], diamond production[4,5], earth sciences[6,7], and so on. Typically, the solid-solid phase transitions can be categorized into two main classes. One is the displacive (martensitic) type, where the atomic configuration evolves with short-range shifts of atoms, and no chemical bonds are broken. The other one is the reconstructive type which involves breaking and reconstruction of

part of the chemical bonds and exhibits drastic changes at the transition point with large latent heat and thermal hysteresis[8]. In contrast to the displacive phase transitions, the dynamics of the reconstructive phase transitions are usually slow because of the need to overcome a large free-energy barrier[9–12]. This raises an important question of whether it is possible to obtain ultrafast dynamics in a reconstructive phase transition, which is highly desirable for designing novel functional devices with fast responses to external stimuli. Interestingly, the

[1]Shenyang National Laboratory for Materials Science, Institute of Metal Research, Chinese Academy of Sciences, Shenyang 110016, China. [2]School of Materials Science and Engineering, University of Science and Technology of China, Shenyang 110016, China. [3]State Key Laboratory of Solidification Processing, International Center for Materials Discovery, School of Materials Science and Engineering, Northwestern Polytechnical University, Xi'an 710072, China. [4]Key Laboratory for Anisotropy and Texture of Materials (Ministry of Education), School of Materials Science and Engineering, Northeastern University, Shenyang 110819, China. [5]University of Vienna, Faculty of Physics and Center for Computational Materials Science, Kolingasse 14-16, A-1090 Vienna, Austria. [6]These authors contributed equally: Mingfeng Liu, Jiantao Wang, Junwei Hu. ✉e-mail: ptliu@imr.ac.cn; haiyang.niu@nwpu.edu.cn

isostructural phase reconstruction from the low-temperature semiconducting $\beta$-Ti$_3$O$_5$ phase to the high-temperature metallic $\lambda$-Ti$_3$O$_5$ phase observed in recent experiments is one representative example meeting such a condition[13–17]. The transition is of first order with a large latent heat and exhibits abnormal ultrafast and reversible characteristics[13–17]. The reversibility can be achieved by applying pressure and heat, pressure and light, or pressure and current[13–15]. These unique structural and functional properties of Ti$_3$O$_5$ therefore have stimulated extensive research interests towards technological applications in optical storage media[13], energy storage[13–15,18–20], solar steam generation[21], and gas sensors[22–24].

To understand the nature of the phase transformation between the $\beta$ and $\lambda$ phases, many studies have been carried out. It was argued that the phase transition is driven by the coupling between the crystal lattice and excited electrons (or holes)[25] or induced by tensile strain[26]. Recent ultrafast powder X-ray diffraction (XRD) measurements, however, demonstrated that the strain waves propagating in a timescale of several picoseconds govern the phase transition[16]. Due to the anisotropy of the strain associated with the phase transition, the photoinduced phase transition in a single crystal of Ti$_3$O$_5$ occurs only when the pump pulse is applied on the *ab* plane[17]. This process has been recently resolved by direction-dependent interface propagation using interface models optimized by machine-learned force field method[27]. A recent study on doping reveals that it primarily alters the relative energy of both phases, but not the activation barrier[28]. Despite extensive studies, the underlying transition mechanism at an atomic level, particularly the kinetic pathway for the ultrafast and reversible transition, remains unclear. This is a known, challenging task for both experimental measurements and theoretical modeling. Specifically, experimental measurements often lack sufficient spatial and temporal resolution to capture atomistic events, whereas theoretical modeling requires accurate sampling of the potential energy surface (PES). Ab initio molecular dynamics (AIMD), albeit with high accuracy, struggle in terms of the accessible time and length scales. Although machine-learning potentials (MLPs)[29–40] can significantly speed up MD simulations while retaining first-principles accuracy, exploring the time scales over which the $\beta$–$\lambda$ phase transition occurs still represents a challenge. By contrast, advanced sampling techniques such as metadynamics[41–43] allow for efficient sampling of the PES of interest and therefore have been successfully used to study barrier-crossing rare events[10–12,44–49].

In this work, we have developed an accurate and efficient MLP through a combined on-the-fly active learning and advanced sampling method. It allows us to perform metadynamics simulations and large-scale MD simulations with ab initio accuracy and discover a kinetically favorable microscopic mechanism for the reconstructive $\beta$–$\lambda$ phase transition. Our results unveil that the phase transition undergoes an interesting in-plane nucleated layer-by-layer kinetic transformation pathway. This is manifested by favorable in-plane intra-cell atomic movements forming two-dimensional (2D) nuclei, followed by propagating to neighboring layers via intermediate metastable crystalline phases comprised of $\beta$-like and $\lambda$-like structural motifs when the lattice strain along the *c* axis increases. Interestingly, we find that superlattices consisting of any combination of $\beta$-like and $\lambda$-like building blocks along the *c* axis are all metastable phases with no imaginary phonon modes. The presence of intermediate layer-by-layer kinetic transformation pathways greatly reduces the free-energy barrier, thereby naturally explaining why an ultrafast and reversible phenomenon can appear in a reconstructive solid-solid phase transition.

## Results
### Machine-learning potential development
Let us start with the MLP construction and validation. To develop an accurate MLP for describing the $\beta$–$\lambda$ phase transition, a representative training dataset covering the phase space of interest is indispensable. This was achieved by combining an on-the-fly active learning

procedure and the enhanced sampling technique (see "Methods"). The final training dataset consists of 3775 structures of 96 atoms, on which the MLP was generated using a moment tensor potential[50]. Through the kernel principal component analysis[51] using the smooth overlap of atomic position descriptors[52], we found that the training structures are indeed very representative and cover a wide range of the phase space. In particular, the phase transition pathway is well sampled by metadynamics simulations (Supplementary Fig. 1a). The generated MLP was carefully validated on a test dataset including 748 structures of 96 atoms, and it is capable of accurately predict the lattice parameters, energy-volume curves, and phonon dispersion relations of both $\beta$- and $\lambda$-Ti$_3$O$_5$, all in good agreement with the underlying density functional theory (DFT) calculations as well as available experimental data (Supplementary Table 1, Supplementary Fig. 1b, c and Supplementary Fig. 2). For more details on the MLP construction and validation, we refer to "Methods". It is worth mentioning that the present MLP is also able to describe well the high-temperature $\alpha$-Ti$_3$O$_5$ phase (Supplementary Table 1 and Supplementary Fig. 1b, c). In particular, the presence of imaginary phonon modes at 0 K and the absence of imaginary phonon modes at high temperatures due to anharmonic phonon-phonon interactions have been well reproduced by our developed MLP (Supplementary Fig. 1). This is not unexpected, since $\alpha$-Ti$_3$O$_5$ shares similar local structural motifs as $\lambda$-Ti$_3$O$_5$.

### Kinetically favorable layer-by-layer transformation pathway
Both the $\beta$- and $\lambda$-Ti$_3$O$_5$ phases have a monoclinic structure with the same space group of *C*2/*m* (Fig. 1d). In addition, their lattice parameters are very close, with the $\lambda$ phase exhibiting a slightly larger *c* lattice constant (Supplementary Table 1). The noticeable difference between the two phases lies in that they exhibit distinct local structural motifs around the Ti3-Ti3 dimers, whereas the local structural environments around Ti1 and Ti2 atoms are similar. The phonon mode analysis indicates that the $A_g$ phonon mode at $\Gamma$ with 3.72 THz is responsible for the transition from $\beta$ to $\lambda$, as the mode softens as the temperature increases. Reversibly, when going from $\lambda$ to $\beta$, the $A_g$ phonon mode at $\Gamma$ with 4.58 THz plays a dominant role. It becomes soft as the pressure is increased over 3 GPa (see Supplementary Fig. 3). We note that the $B_u$ mode at $\Gamma$ previously identified in ref. 14 is not the decisive phonon mode to derive the phase transformation, since by following the eigenvector of this mode, one can not obtain the transition state with the typical feature of the rotation of Ti3-Ti3 dimers. Our phonon mode analysis is in line with our metadynamics simulations (see "Methods"), which show that during the $\beta$ to $\lambda$ transition, the Ti3-Ti3 dimers undergo a gradual rotation, accompanied by the breaking of Ti3-O4 bond and the formation of Ti3-O5 bond (Supplementary Fig. 4). It is interesting to note that only the atoms involving the Ti3-Ti3 dimers and associated O atoms participate in the intra-cell atomic movements, while the other atomic layers including Ti1 and Ti2 atoms almost remain unchanged. Our observations corroborate the structural changes inferred by femtosecond powder XRD[16].

On the thermodynamic side, we computed Gibbs free energy via metadynamics simulations. For the choice of collective variable (CV), following ref. 46 the intensity of the XRD peak at $2\theta = 22°$ [corresponding to the (100) plane] was used (Fig. 1a). It turns out that the employed CV is very efficient and can effectively distinguish the two phases, as manifested by frequent reversible transitions between the two phases (Fig. 1b). Using the reweighting technique[53], the Gibbs free energies of the two phases were computed. By fully accounting for the anharmonic effects, the calculated phase transition temperature $T_c$ and phase transition enthalpy $\Delta H$ at 0 GPa are 535 K and 0.091 eV/f.u., respectively, in good agreement with the experimental values ($T_c = 470$ K and $\Delta H = 0.124 \pm 0.01$ eV/f.u.)[14] (Fig. 1c). By contrast, the quasi-harmonic approximation significantly overestimates the transition temperature (Fig. 1c). Using a similar procedure, a full ab initio pressure-temperature phase diagram has been established (Fig. 1d).

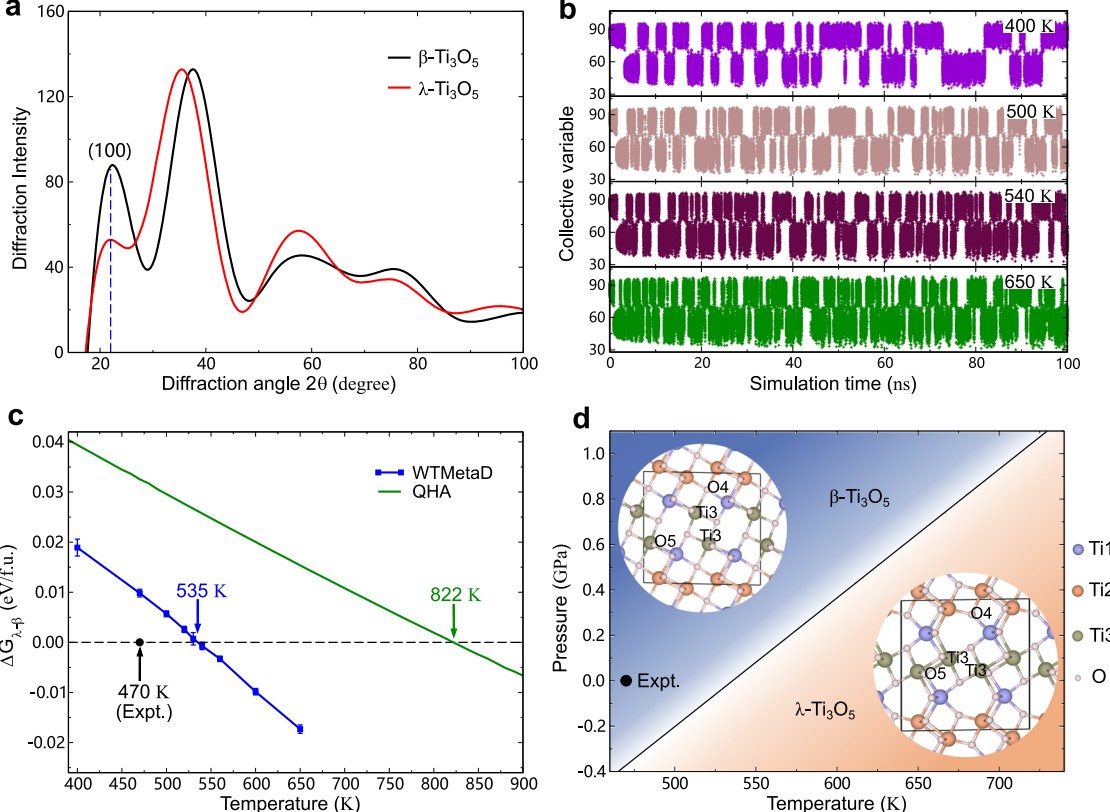

**Fig. 1 | Metadynamics simulations and phase diagram. a** Simulated XRDs of $\beta$- and $\lambda$-$Ti_3O_5$ using a 96-atom supercell. **b** Time evolution of collective variable at different temperatures and 0 GPa. **c** Free-energy difference $\Delta G_{\lambda-\beta}$ calculated by the well-tempered variant of metadynamics (WTMetaD) and the quasi-harmonic approximation (QHA) as a function of temperature. The energy is given in electron volt per formula unit (eV/f.u.). The error bar indicates the standard deviation from three independent simulations. The predicted and experimental phase transition temperatures are indicated by arrows. **d** Pressure-temperature phase diagram predicted by metadynamics simulations. Insets show the crystal structures of $\beta$- and $\lambda$-$Ti_3O_5$, where the black lines indicate the unit cell. The experimental data (denoted as Expt.) in **c**, **d** are taken from ref. 14. Source data are provided as a Source Data file. The structure models used are provided in Supplementary Data 1.

Kinetically, the direct phase transformation from $\beta$ to $\lambda$ in a concerted manner needs to overcome a larger energy barrier of 0.25 eV/ f.u., as revealed by our variable-cell climbing image nudged elastic band (CI-NEB) calculations at 0 K (see Supplementary Fig. 5 and also ref. 54). This seems incompatible with the experimentally observed ultrafast and reversible nature of the transition[14,16], indicating that the phase transition must undergo intermediate barrier-lowering kinetic transformation pathways. To confirm this, we computed the PES through metadynamics simulations using a supercell including three Ti3-Ti3 layers (Fig. 2a). The CVs were designed to be able to well describe the degree of the rotations of Ti3-Ti3 dimers in each layer (see "Methods" and Supplementary Fig. 6a). The computed free energy surface at 600 K and 0 GPa as a function of the employed CVs indeed identifies four energy minima, which correspond to $\beta$-$Ti_3O_5$, a $\beta\beta\lambda$- stacking phase, a $\beta\lambda\lambda$-stacking phase, and $\lambda$-$Ti_3O_5$, respectively (Fig. 2b). Our phonon calculations revealed that both the $\beta\beta\lambda$-stacking and $\beta\lambda\lambda$-stacking phases are dynamically stable (Supplementary Fig. 8). More interestingly, we predicted that all superlattices consisting of any combination of $\beta$-like and $\lambda$-like structural motifs along the $c$ direction are dynamically stable (Supplementary Fig. 7 and Supplementary Fig. 8). They all adopt the same space group of $C2/m$. The dynamical stabilities were confirmed by performing phonon calculations on superlattices with up to eight building blocks using the developed MLP. The MLP predictions were verified by DFT calculations on small superlattices such as two-layer-stacking $\beta\lambda$ phase and three-layer- stacking $\beta\beta\lambda$ and $\beta\lambda\lambda$ phases (Supplementary Fig. 8a–c).

We further computed the energy-volume curves at 0 K and 0 GPa (Fig. 2c). One can see that the system volume (mostly the $c$ lattice

constant) increases as the number of $\lambda$-like building blocks increases. The energy-volume curves of the metastable $\beta\beta\lambda$ and $\beta\lambda\lambda$ phases are evenly located between the ones of $\beta$-$Ti_3O_5$ and $\lambda$-$Ti_3O_5$. They almost share a common tangent line, implying that under specific negative pressure, the $\beta$ to $\lambda$ phase transformation is likely to go through the intermediate $\beta\beta\lambda$ and $\beta\lambda\lambda$ metastable phases. Indeed, the free energies computed by metadynamics simulations at 600 K and 0 GPa demonstrated that as compared to the concerted $\beta-\lambda$ phase transition, the existence of the intermediate $\beta\beta\lambda$ and $\beta\lambda\lambda$ metastable phases significantly reduces the energy barrier from 0.19 eV/f.u. to 0.07 eV/f.u. (Fig. 2d), a small value for a solid-solid phase transition. The metadynamics free energy calculations are in line with the variable-cell CI-NEB calculations using the same three-layer supercell (Supplementary Fig. 9a). We would like to stress that the obtained layer-by-layer kinetic transformation mechanism is not limited to the three-layer supercell. Similar observations were also obtained for a larger five-layer supercell (Supplementary Fig. 9b). We note that this is the so-called Ostwald's "rule of stages"[55], which has been experimentally proved in solid-solid transitions of colloidal crystals mediated by a transient liquid intermediate[56]. However, the finding of metastable crystalline phases serving as the intermediate for a reconstructive solid-solid phase transition is rare.

The discovered kinetically favorable layer-by-layer transformation pathway is essentially determined by the specific structure correlation between the two phases. The shared common structural features (i.e., non-displacive Ti1-Ti1 and Ti2-Ti2 layers) exhibit twofold effects. One is to connect the $\beta$- or $\lambda$-like structural building blocks resulting in Lego- like metastable phases, and the other is to serve as blocking layers for

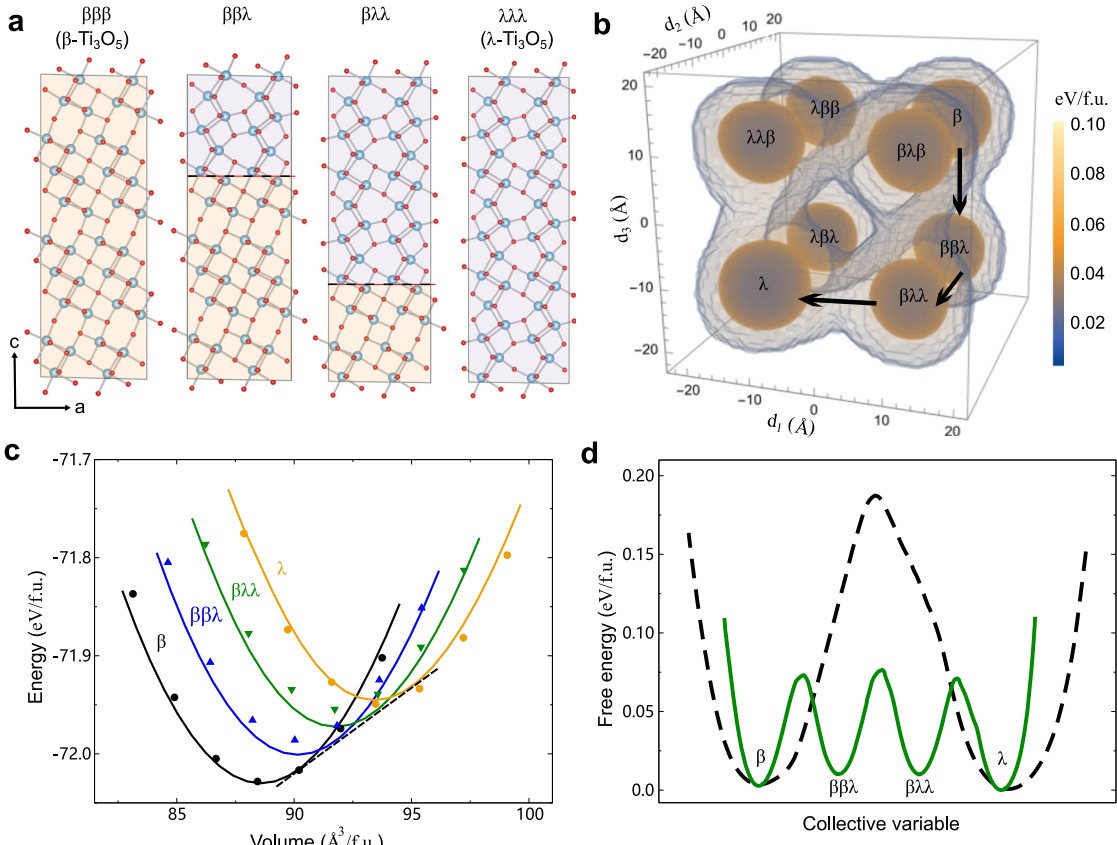

**Fig. 2 | Predicted metastable phases and their potential energy surface.**
**a** Structures of $\beta$-Ti$_3$O$_5$, $\beta\beta\lambda$-stacking metastable phase, $\beta\lambda\lambda$-stacking metastable phase and $\lambda$-Ti$_3$O$_5$, respectively. The light yellow and purple colors represent the $\beta$-like and $\lambda$-like structural motifs, respectively. The blue and red balls represent Ti and O atoms, respectively. **b** Free-energy surface (in electron volt per formula unit, i.e., eV/f.u.) computed from metadynamics simulations at 600 K and 0 GPa. For definition of collective variables $d_i$ ($i$ = 1, 2 and 3), we refer to "Methods" and Supplementary Fig. 6a. Note that $\beta\beta\lambda$, $\lambda\beta\beta$, and $\beta\lambda\beta$ are symmetry-equivalent structures,

and similarly, $\beta\lambda\lambda$, $\lambda\beta\lambda$, and $\lambda\lambda\beta$ are symmetry-equivalent structures. **c**, Energy-volume curves predicted by DFT (circles and triangles) and machine-learning potential (solid lines) at 0 K and 0 GPa. **d** Potential energy surface (PES) obtained at 600 K and 0 GPa as a function of employed collective variable. The black dashed line denotes the PES for a concerted $\beta$–$\lambda$ phase transition, whereas the green solid lines denotes the PES for the transition pathway through intermediate metastable phases (corresponding to arrows in **b**). Source data are provided as a Source Data file. The structure models used are provided in Supplementary Data 1.

the inter-layer propagation leading to the layer-by-layer kinetic transformation pathway. During the $\beta$–$\lambda$ phase reconstruction, only Ti and O atoms associated with the Ti3-Ti3 layers are involved. In addition, only the relatively weak Ti3-O4 and Ti3-O5 chemical bonds need to be broken for $\beta{\rightarrow}\lambda$ and $\lambda{\rightarrow}\beta$, respectively. This is evidenced by the DFT calculated electron localization functions (Supplementary Fig. 10) and also by the chemical analysis using integrated crystal-orbital Hamiltonian populations by Fu et al.[25]. In contrast, the Ti3-Ti3 chemical bonds forming a $\sigma$-bonding of $d_{xy}$-like states[21] remain intact. This explains why the value of the energy barrier for the $\beta$–$\lambda$ transformation is low. Our observation provides a natural explanation for the ultrafast and reversible nature of the $\beta$–$\lambda$ transition[14,16].

**Evidence of in-plane nucleated multistep kinetic mechanism**
To further corroborate the layer-by-layer multistep kinetic mechanism for the $\beta$–$\lambda$ phase transition as revealed by metadynamics simulations, we performed unbiased direct MD simulations. However, our tests showed that direct MD simulations were not capable of overcoming the energy barrier for the $\beta$–$\lambda$ phase transition on a nanosecond timescale. Nevertheless, applying unidirectional tensile strain along the $c$-axis can significantly reduce the energy barrier (see Supplementary Fig. 11), making direct MD simulations possible[26]. Indeed, with an unidirectional tensile strain in the $c$-direction, the first-order phase transition from $\beta$ to $\lambda$ was successfully reproduced within ~1 ns by direct MD simulations on a 96-atom cell using the MLP, as manifested

by an abrupt jump in the potential energy as well as stress (see Supplementary Fig. 12a, b). A closer inspection of local structure changes from the direct MD simulations verifies the metadynamics results(compare Supplementary Fig. 12c to Supplementary Fig. 4). We note in passing that applying a $c$-axis unidirectional strain in DFT also drives the $\beta$–$\lambda$ phase transition (Supplementary Fig. 13).

In order to exclude the size effects and further clarify the phase transition details, we carried out large-scale MD simulations under continuously increasing tensile strain along the $c$ axis using a large cell with 165,888 atoms (see "Methods" for details). Structural evolutions similarly to the small cell discussed above were observed (compare Fig. 3 to Supplementary Fig. 12). Nevertheless, the 2D in-plane nucleation and growth behavior accompanied by step-wise changes in the potential energy and stress can be better visualized in the large-scale MD simulations (see Fig. 3 and Supplementary Movie 1). It is interesting to observe that under continuously increasing tensile strain the $\beta$ phase does not transform to the $\lambda$ phase simultaneously in one step, but rather the transformation starts by the formation of 2D nuclei in the $ab$-plane and then grows successively layer-by-layer along the $c$ axis mediated by metastable phases (see Fig. 3c). The easy propagation along the $c$ axis arises from the lower energy barrier as compared to other two ($a$ and $b$) directions, as recently revealed by Jütten and Bredow[27]. The deformation involving the Ti3-Ti3 dimers and associated O atoms does not propagate to the neighboring layers until the present layer completes the transition to the $\lambda$-Ti$_3$O$_5$-like structural motifs.

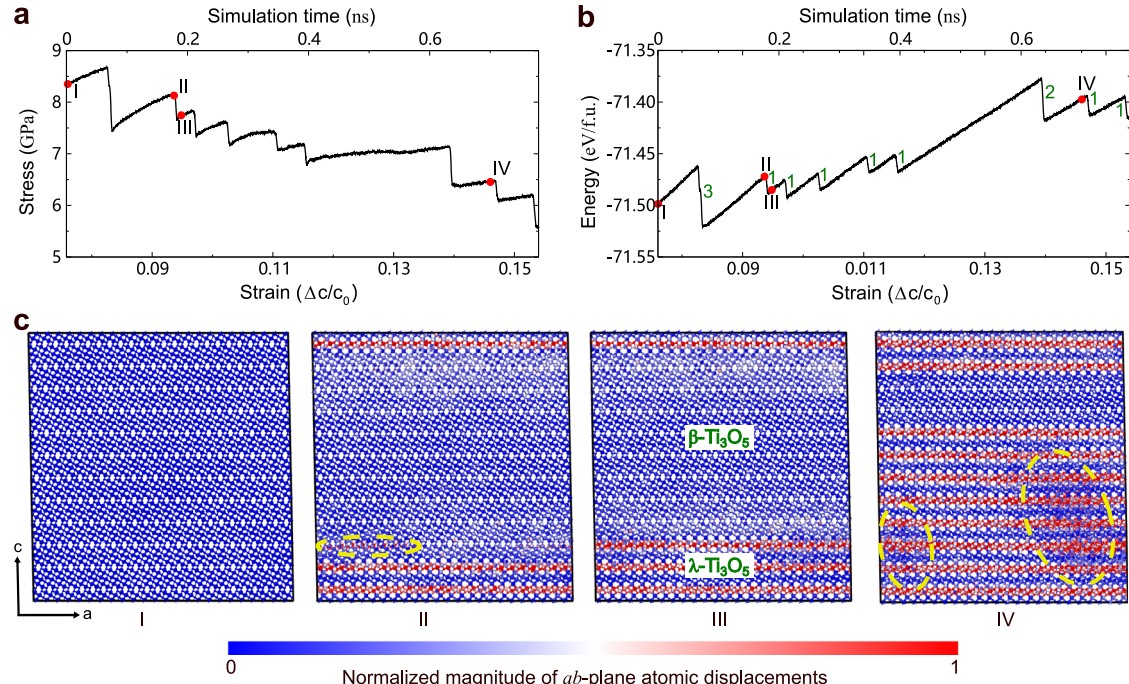

**Fig. 3 | Direct MD simulations under tensile strain at 300 K and 0 GPa using a large cell including 165,888 atoms. a, b** Evolution of stress and potential energy (in electron volt per formula unit, i.e., eV/f.u.) with respect to strain under $c$-axis unidirectional continuous tensile strain (strain rate $10^8$/s). The values close to each jump of the potential energy in **b** indicate the number of layers undergoing the $\beta$ to $\lambda$ transition. **c**, Snapshots at the simulation times indicated by red dots in **a**, **b**. The Ti and O atoms are denoted by large and small balls, respectively. The yellow dashed circle in snapshot II highlights the 2D nucleation in the $ab$-plane, while the two yellow dashed circles in snapshot IV mark the defective regions. Source data are provided as a Source Data file. The configurations presented are provided in Supplementary Data 1.

The characteristic 2D growth behavior of the $\beta$- to $\lambda$-Ti$_3$O$_5$ transition is reflected by the step-wise potential energy as a function of strain (Fig. 3b). The growth of the $\lambda$-like structural motifs causes the tensile strain for the neighboring layers. This would decrease the kinetic energy barrier (Supplementary Fig. 11), thereby facilitating the layer-by-layer $\beta$ to $\lambda$ transformation. We note that the 2D layer-by-layer growth behavior can be better captured by using a smaller strain rate (Supplementary Fig. 14). In line with previous metadynamics results, our large-scale unbiased MD simulations consolidate the in-plane nucleated layer-by-layer kinetic mechanism for the $\beta$ to $\lambda$ transition and corroborate the strain wave pathway derived by experiment[16]. We notice that during the $\beta$ to $\lambda$ transition in the large-scale MD simulations, a defective metastable phase featured by incomplete rotation of Ti3-Ti3 dimers appears (Fig. 3c and Supplementary Fig. 15). This is likely caused by anisotropic strain energies associated with inhomogeneous lattice distortions[17], because this metastable phase is found to be stable only under tensile strain and will transform to the more stable $\beta$ phase after full structural relaxation.

It should be noted that the 2D in-plane nucleation and growth behavior of the $\beta$–$\lambda$ phase transition is robust, regardless of the presence of external tensile strain. To demonstrate this, we carried out a computational experiment by artificially embedding a $\lambda$-Ti$_3$O$_5$-like nucleus with a radius of 60 Å in the $\beta$-Ti$_3$O$_5$ matrix (see "Methods" for details). The employed supercell model including 122,880 atoms is shown in Fig. 4a, b. This allows us to overcome the energy barrier of the $\beta$ to $\lambda$ transition and study the propagation and annihilation processes of the $\lambda$-Ti$_3$O$_5$-like nucleus by just tuning the temperature and pressure in a direct isothermal-isobaric MD simulation without applying any external forces. We found that at ambient pressure the preexisting $\lambda$-Ti$_3$O$_5$-like nucleus grows only within the layer and eventually extends to the full layer as the temperature goes over 1080 K. Below 1080 K it gradually disappears (see Fig. 4c, d and Supplementary Movie 2). However, the inter-layer propagation did not take place, in contrast to

the tensile MD simulations. This is due to the insufficient driving force along the $c$ axis. We note that the temperature at which the preexisting $\lambda$-Ti$_3$O$_5$-like nucleus starts to grow is higher than the thermodynamic phase transition temperature. This is likely caused by the presence of compressive strain in the nucleus. Indeed, the critical transition temperature is decreased when reducing the pressure to a negative value, and vice versa (Supplementary Fig. 16), in accordance with the established pressure-temperature phase diagram (Fig. 1). Our computational experiments combined with the $c$-axis tensile MD simulations clearly demonstrate that the intra-layer transformation and growth are more favorable, whereas the inter-layer growth starts to occur only when the $c$-axis strain is sufficiently large.

### Kinetics of in-plane nucleation and growth

As discussed above, the $\beta$–$\lambda$ phase transformation is initiated through in-plane nucleation and subsequently expands in a layer-by-layer manner. Nevertheless, further investigation is required to understand the specific kinetics involved in the process of in-plane nucleation and growth. To this end, we carried out variable-cell CI-NEB calculations using the MLP for transforming one layer of $\beta$-Ti$_3$O$_5$ from $\beta$-like to $\lambda$-like structural motifs. The results are displayed in Fig. 5. It is clear that the direct transformation in a concerted manner is energetically unfavorable because of the need to overcome a relatively large energy barrier (0.09 eV/f.u.) (see Path I in Fig. 5b). By contrast, the transformation via the in-plane nucleation is kinetically more favorable, with a decreased energy barrier of 0.07 eV/f.u. for the employed specific supercell (see Path II in Fig. 5b). We note that for the case of the nucleation mechanism, the energy barrier $B$ is independent of the system size $N$, resulting in $B/N \sim 0$ when $N$ is large, while for the case of a collective mechanism, the energy barrier $B$ increases linearly with the system size $N$, leading to a constant ratio $B/N$ with respect to $N$ (see Supplementary Fig. 18). Furthermore, one can see that once the nucleation occurs, the subsequent in-plane

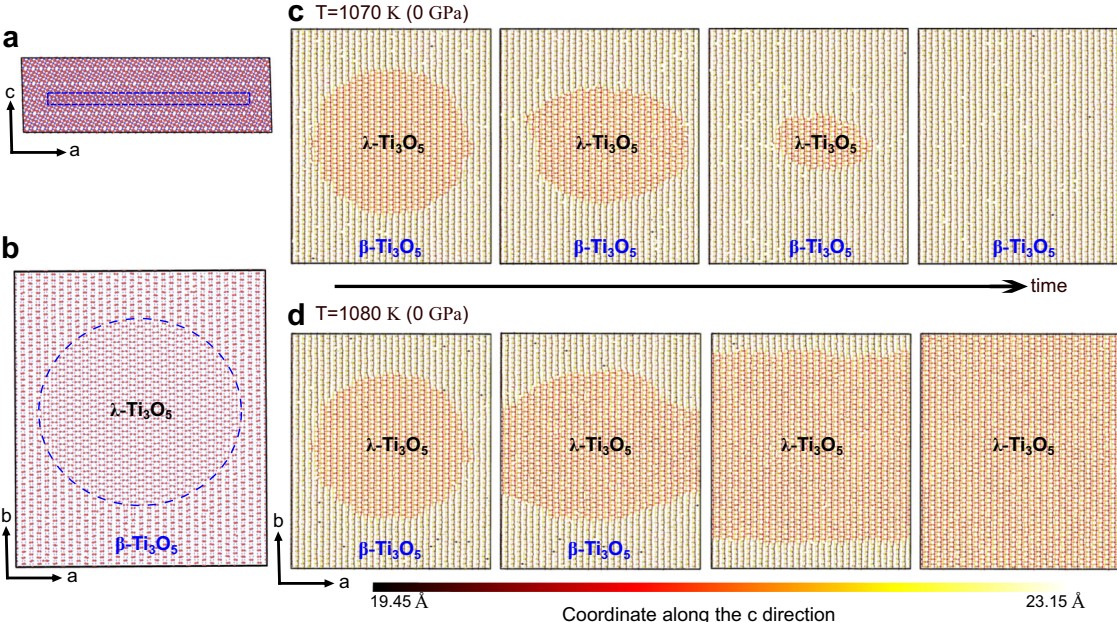

**Fig. 4 | Large-scale 2D phase growth MD simulations. a, b** Side and top views of employed structural model consisting of 122,880 atoms. A $\lambda$-Ti$_3$O$_5$-like nucleus with a thickness equal to the $c$ axis lattice constant was artificially created in a region with a radius of 60 Å (marked by dashed lines) in the middle of the $\beta$-Ti$_3$O$_5$ matrix. The large and small spheres represent the Ti and O atoms, respectively. **c** Snapshots from an MD simulation at 1070 K and 0 GPa. **d** Snapshots from an MD simulation at 1080 K and 0 GPa. Note that in **b–d** only the layer containing the $\lambda$-Ti$_3$O$_5$-like nucleus is shown. The initial and final configurations are provided in Supplementary Data 1.

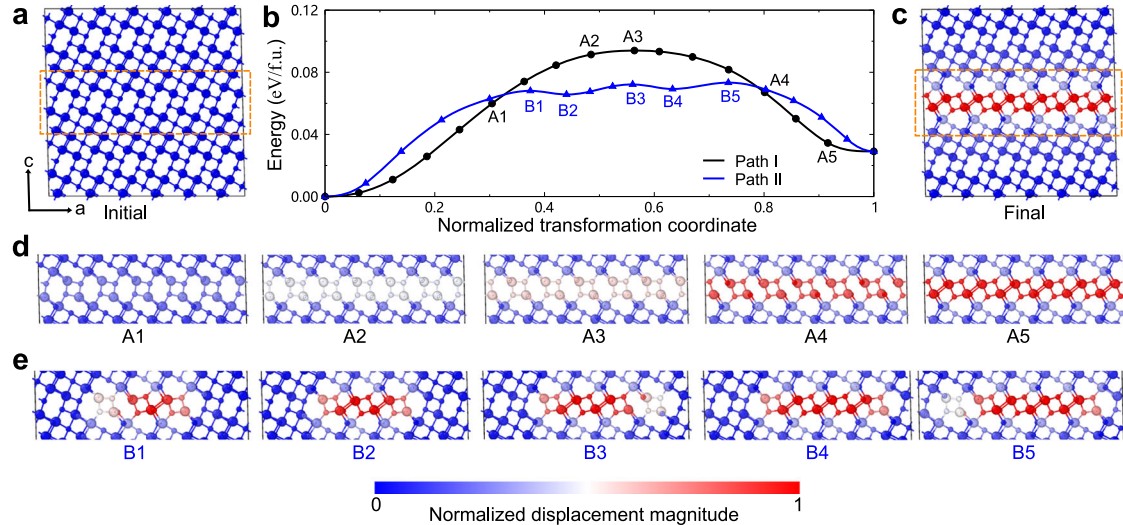

**Fig. 5 | Variable-cell climbing image nudged elastic band calculations for transforming one layer of $\beta$-Ti$_3$O$_5$ from $\beta$-like to $\lambda$-like structural motifs. a** Employed initial structure of $\beta$-Ti$_3$O$_5$ including 288 atoms. **b** Computed relative energies (in electron volt per formula unit, i.e., eV/f.u.) as a function of normalized transformation coordinate for two pathways. Path I denotes that the transformation proceeds in a concerted manner, whereas Path II indicates that the transformation initiates from nucleation and then proceeds via intermediate phases. **c** Final structure of the transformation. **d** Selected structures along Path I. **e** Selected structures along Path II. Note that for a better presentation, only the atoms inside the dashed rectangle in **a** are shown, and the atoms are color-coded according to the magnitude of displacements relative to the initial $\beta$-Ti$_3$O$_5$ phase. The large and small balls represent the Ti and O atoms, respectively. Source data are provided as a Source Data file. The structure models used are provided in Supplementary Data 1.

growth proceeds with ease. Intriguingly, the in-plane growth process unveiled the presence of further intermediate metastable phases (B2 and B4 in Fig. 5b), whose dynamical stabilities were confirmed by phonon calculations (see Supplementary Fig. 17). The transformation from the metastable phase B2 to the metastable phase B4 only needs to overcome a minute energy barrier of 6 meV/f.u. (see Fig. 5b),

indicating the easy in-plane growth. More detailed transformations for Paths I and II are provided in Supplementary Movie 3.

## Discussion
Accurate atomistic simulations of solid-solid structural phase transitions are challenging. Simulations are hindered by the slow dynamics

and the lack of accurate and efficient interatomic potentials. Here, the issue has been addressed by developing an efficient MLP with near-first-principles accuracy through a combination of an on-the-fly active learning method and an advanced sampling technique. It is remarkable that the generated MLP based on a simple standard PBE functional is capable of simultaneously describing well the lattice parameters, energy-volume curves, phase transition temperature, and phonon dispersion relations of both the $\beta$- and $\lambda$-Ti$_3$O$_5$ phases, and the high-temperature $\alpha$-Ti$_3$O$_5$ phase. It also predicts Lego-like metastable phases that are not included in the training dataset. It is noteworthy that the PBE functional incorrectly predicts $\beta$-Ti$_3$O$_5$ to be a metal. PBEsol, SCAN, and hybrid functionals are also unable to open a band gap in $\beta$-Ti$_3$O$_5$ if magnetic ordering is not considered. Our study indicates that the electron correlation effects may not play a decisive role in the $\beta$–$\lambda$ structural phase transition. Fundamentally, the implications of the present study are intriguing as it suggests that the phase transitions in Ti$_3$O$_5$ can be described on the level of the simplest mean field theory neglecting magnetism. It might be worthwhile to study the influence of vibrational contributions on phase transitions in other materials that are broadly considered to be strongly correlated. We note in passing that very recently an MLP was also developed by Jütten and Bredow[27] to study the pressure effect on the phase transition of Ti$_3$O$_5$. However, their developed MLP was mainly used to optimize the interface models with mixed phases, while the kinetic phase transition details have not been explored.

The carefully validated MLP not only allows us to perform meta-dynamics simulations, but also enables us to carry out large-scale MD simulations. Using the intensity of the XRD peak as a collective variable for the metadynamics simulations, a full ab initio thermodynamic pressure-temperature phase diagram for the $\beta$ and $\lambda$ phases has been constructed, in good agreement with experiments. By performing large-scale MD simulations under $c$ axis unidirectional tensile strain and using computational experiments with a preexisting $\lambda$-like nucleus in a $\beta$ matrix, we have clarified that the phase transformation starts with favorable 2D nucleation in the $ab$ plane, and then propagates to the neighboring layers through a multistep barrier-lowering kinetic pathway via intermediate metastable crystalline phases. This kinetically favorable in-plane nucleated multistep mechanism has not been revealed before and provides a straightforward interpretation of the experimentally observed unusual ultrafast dynamics in the reconstructive $\beta$–$\lambda$ transition. Due to the structural similarity, we expect that the microscopic transition mechanism obtained here should also apply to phase transitions of other titanium suboxides, e.g., the $\gamma$- to $\delta$-Ti$_3$O$_5$ transition[57] and phase transitions in Ti$_4$O$_7$[58].

Finally, we would like to emphasize that the observed metastable phases are particularly interesting, since periodic superlattices formed by any combination of $\beta$-like and $\lambda$-like building blocks along the $c$ axis are all dynamically stable. This flexibility opens avenues for tuning functional properties, e.g., insulator-to-metal transition and specific heat modulation. Consequently, our findings hold great potential for practical applications in the design of novel energy storage media, optical storage media, and sensor devices. Moreover, given the success of the developed MLP in predicting many undiscovered metastable phases, combining the present MLP with structure search tools would allow us to efficiently explore more unknown stable/metastable phases, not just limited to a small number of atoms[59]. Our work not only gives important insight into the microscopic mechanism for the $\beta$–$\lambda$ phase transition, but also provides useful strategies and methods that can be widely used to tackle other complex structural phase transitions.

## Methods
### First-principles calculations
The first-principles calculations were performed using the Vienna ab initio simulation package (VASP)[60]. The generalized gradient approximation of Perdew-Burke-Ernzerhof (PBE)[61] was used for the exchange-correlation functional, which yields an excellent description of the experimental lattice parameters of both $\beta$- and $\lambda$-Ti$_3$O$_5$ as compared to other density functionals such as PBEsol[62] and SCAN[63] (see Supplementary Table 1). The standard projector-augmented-wave potentials (Ti_sv and O) were used. A plane wave cutoff of 520 eV and a $\Gamma$-centered $k$-point grid with a spacing of 0.21 Å$^{-1}$ between $k$ points (corresponding to a $3 \times 3 \times 3$ $k$-point grid for a 96-atom supercell) were employed. This ensures that the total energy is converged to better than 1 meV/atom. The Gaussian smearing method with a smearing width of 0.05 eV was used. Whenever ground state structures were required, the electronic optimization was performed until the total energy difference between two iterations was less than 10$^{-6}$ eV. The structures were optimized until the forces were smaller than 1 meV/Å. Since it was experimentally found that the $\beta$ phase is a nonmagnetic semiconductor, while the $\lambda$ phase is a weak Pauli paramagnetic metal[13,64], nonmagnetic setups were thus adopted for all the calculations. This is a reasonable choice, since adopting the PBE functional with nonmagnetic setups yields good descriptions of not only the electronic structures[13,21,64], but also the structural and thermodynamical properties of both $\beta$ and $\lambda$ phases (See Supplementary Note 1 for more detailed discussions). The D3 dispersion corrections[65] were not included, since, for strongly ionic bulk materials, the D3 corrections are not expected to be accurate, and we found that inclusion of D3 correction deteriorates the description of lattice parameters and energy differences between the two phases (See Supplementary Table 1). The harmonic phonon dispersions were calculated by finite displacements using the Phonopy code[66]. The anharmonic phonon dispersions at finite temperatures were obtained using the temperature-dependent effective potential method[67] and the anharmonic force constants were extracted using the ALAMODE code[68]. The variable-cell climbing image nudged elastic band method[69] was used to estimate the energy barrier for the $\beta$–$\lambda$ phase transition.

### MLP construction and validation
The training dataset was initially constructed using the on-the-fly active learning method based on the Bayesian linear regression[70,71] using the separable descriptors[72], which allows us to efficiently sample the phase space and automatically collect the representative training structures during the AIMD simulations. The cutoff radius for the three-body descriptors and the width of the Gaussian functions used for broadening the atomic distributions of the three-body descriptors were set to 6 Å and 0.4 Å, respectively. The number of radial basis functions and maximum three-body momentum quantum number of the spherical harmonics used to expand the atomic distribution for the three-body descriptors were set to 14 and 4, respectively. The parameters for the two-body descriptors were the same as those for the three-body descriptors. The AIMD simulations were performed by heating the $\beta$- and $\lambda$-Ti$_3$O$_5$ phases of 96-atom supercell from 100 to 1500 K at ambient pressure and 10 GPa, in an isothermal-isobaric ensemble using a Langevin thermostat[73] combined with the Parrinello-Rahman method[74,75]. Eventually, 3277 structures were selected in this step.

Since the phase transition from $\beta$- to $\lambda$-Ti$_3$O$_5$ is a barrier-crossing process, unbiased MD simulations hardly sample the PES along the transition path, even using the MLP at high temperatures. To address this issue, we first refitted the on-the-fly generated dataset using a moment tensor potential (MTP)[50] that is in general one order of magnitude faster than the kernel-based methods for a comparable accuracy[76,77]. For the MTP potential training, a cutoff radius of 6.0 Å was used, and the radial basis size was set to 8. The MTP basis functions were selected such that the level of scalar basis $B_\alpha$ is less than or equal to 26 (i.e., lev$B_\alpha \leq 26$). The weights expressing the importance of energies, forces, and stress tensors were set to 1.0, 0.05, and 0.05, respectively. The regression coefficients (in total, 2153) were obtained

by a non-linear least square optimization using the Broyden-Fletcher-Goldfarb-Shanno algorithm[78]. Employing the generated MTP, we performed a long-timescale metadynamics simulations at 600 K and 0 GPa on a 96-atom supercell using the intensity of the XRD peak at $2\theta = 22°$ as a collective variable following ref. 46 (see Fig. 1a). Afterwards, we computed the extrapolation grade for the structures from the metadynamics trajectory according to the D-optimality criterion[79]. The structures with an extrapolation grade over 3.0 (in total, 498 structures) were selected and added to the on-the-fly generated dataset. In the end, the training dataset contained 3775 structures of 96 atoms, on which the final MTP potential was trained using the MLIP package[80].

The MTP potential was validated on a test dataset containing 748 structures of 96 atoms, which were randomly chosen from a metadynamics trajectory at 600 K using the generated MTP potential. These test structures cover a fraction of the phase space represented by the training dataset, as illustrated by the kernel principal component analysis in Supplementary Fig. 2a. The training and validation root-mean-square-errors (RMSEs) on energies, forces, and stress tensors calculated by the MTP are illustrated in Supplementary Fig. 2, showing a high accuracy of the generated MTP. Moreover, the accuracy of the MTP was carefully validated by predicting the lattice parameters, energy-volume curves, and phonon dispersion relations of both $\beta$- and $\lambda$-$Ti_3O_5$, exhibiting excellent agreement with the underlying PBE results as well as available experimental data (see Supplementary Fig. 1b, c and Supplementary Table 1).

**Metadynamics simulations**

In order to simulate the $\beta–\lambda$ phase transition and calculate the thermodynamic phase diagram, metadynamics simulations at various temperatures and pressures were carried out using the WTMetaD[42]. The isothermal-isobaric WTMetaD simulations were carried out using the LAMMPS code[81] patched with the PLUMED code[82]. The temperature was controlled using the stochastic velocity rescaling thermostat[83] with a relaxation time 0.1 ps. The pressure was controlled with the Parrinello-Rahman barostat[75] with a relaxation time of 10 ps. The equations of motion were integrated using a time step of 2 fs. The bias factor that characterizes the rate of change of the deposited Gaussian height was set to 20. Gaussians with a width of 1 CV unit and an initial height of 5 kJ/mol were deposited every 1 ps to construct the history-dependent bias potential. For the choice of collective variable, the intensity of the XRD peak at $2\theta = 22°$ was used, following ref. 46 (Fig. 1a). The Gibbs free energies were computed using the reweighting technique[53].

For the description of layer-by-layer growth, we employed the summation of the projected displacements of all the Ti3-Ti3 dimers along the $a$ and $c$ axes in each layer as the CVs, which are able to describe the degree of rotations of Ti3-Ti3 dimers in each layer and well distinguish the four phases (i.e., $\beta$, $\beta\beta\lambda$, $\beta\lambda\lambda$, and $\lambda$) (see Supplementary Fig. 6a for details). When calculating the PES for a concerted $\beta–\lambda$ phase transition in a three-layer structure, the two CVs of the second layer $d_2$ and the third layer $d_3$ were constrained to be equal to that of the first layer $d_1$ so that all the three layers were forced to transform simultaneously from $\beta$ to $\lambda$. For the WTMetaD simulations, the width and the initial height of Gaussian were set to 1.2 Å and 25 kJ/mol, respectively. The bias factor was set to 30. The time interval for the deposit of Gaussians was set to 0.5 ps. The simulation time was set as 200 ns to ensure the convergence. After the sampling was finished, we reweighted the data to obtain the desired free energy surface using an on-the-fly strategy[84].

**Molecular dynamics simulations**

MD simulations were performed using the LAMMPS code[81] in an isothermal-isobaric ensemble. The temperature was controlled using a Nosé-Hoover thermostat[85–87] with a relaxation time of 0.2 ps. The

pressure was controlled with the Parrinello-Rahman barostat[75] with a relaxation time of 0.1 ps. For the $c$-axis unidirectional continuous tensile MD simulations, a supercell consisting of $12 \times 36 \times 12$ 32-atom conventional cells (in total 165,888 atoms) was used. The strain rate was set to be $10^8$/s. To model the intra-layer 2D growth behavior from $\beta$- to $\lambda$-$Ti_3O_5$, we built a $\beta$-$Ti_3O_5$ supercell consisting of $16 \times 48 \times 5$ 32-atom conventional cells (in total, 122,880 atoms) and then artificially created a $\lambda$-$Ti_3O_5$ layer (with a thickness equal to the $c$-axis lattice constant of 32-atom conventional cell) in a region with a radius of 60 Å in the middle of the supercell (see Fig. 4a, b).

## Data availability

The data, including the crystal structures and phonon dispersions of all predicted stable and metastable phases, training and validation datasets, and the developed MLP, have been deposited in the figshare repository (doi: 10.6084/m9.figshare.24279616) (ref. 88). The structure models used are provided in Supplementary Data 1. The other data that support this study's findings are available from the corresponding author upon request. Source data are provided with this paper.

## Code availability

VASP can be acquired from the VASP Software GmbH (see www.vasp.at); MLIP is available at mlip.skoltech.ru; LAMMPS is available at www.lammps.org; PLUMED is available at www.plumed.org; OVITO is available at www.ovito.org; ALAMODE is available at alamode.readthedocs.io; Phonopy is available at phonopy.github.io/phonopy.

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

## Acknowledgements

The work at the Institute of Metal Research is supported by the National Natural Science Foundation of China (Grant no. 52201030 and Grant no. 52188101), the National Key R& D Program of China 2021YFB3501503, Chinese Academy of Sciences (Grant no. ZDRW-CN-2021-2-5), and the National Science Fund for Distinguished Young Scholars (Grant no. 51725103). The work at Northeastern University is supported by the National Natural Science Foundation of China (Grant no. 52331001). J.H. and H.N. acknowledge the support from the National Natural Science Foundation of China (Grant no. 92370118) and the Research Fund of the State Key Laboratory of Solidification Processing (NPU), China (grant no. 2020-QZ-03). G.K. acknowledges the support from the Austrian Science Fund (FWF) within the SFB TACO (Project no. F 81-N).

## Author contributions

P.L. conceived the project. P.L. designed the research with the help of H.N., X.-Q.C. and L.Z. M.L., J.W., P. L., J.H. and H.N. performed the calculations. X.-Q.C., G.K. and L.Z. supervised the project. X.Y., J.L., H.Y., B.Y., Y.S. and C.C. participated in discussions. P.L. wrote the manuscript with inputs from other authors. All authors comment on the manuscript. M.L., J.W. and J.H. contributed equally to this work.

## Competing interests

The authors declare no competing interests.
