## [Peer Review File · Nature Communications]

Layer-by-layer phase transformation in Ti₃O₅ revealed by machine learning molecular dynamics simulationsEditorial Note: Parts of this Peer Review File have been redacted as indicated to remove third-party material where no permission to publish could be obtained.

Reviewer #1 (Remarks to the Author):

The work is interesting, original and timely. The work appears technically correct and the results are likely reliable and interesting for the solid state physics and materials science community. However, I see a problem with the interpretation of the results. The authors say on p. 13 "By contrast, the transformation via the in-plane nucleation is energetically more favorable, with a decreased energy barrier of 0.07 eV/f.u. (see Path II in Fig. 5b)." I have the following comments to this statement.

1 In order to claim that a real nucleation event is observed, one has to show that the barrier B is independent on the system size, which was not the case here. It would be necessary to perform several calculations for larger system sizes.

2 Dividing the barrier B by the number of formula units N and reporting energy per formula unit B/N makes sense in case of collective mechanism when B/N stays constant for increasing N , not in the case of nucleation. If the barrier indeed scales linearly with the system size and B/N stays constant, then what is observed is not nucleation. If instead a real nucleation is observed, with a finite barrier B , by increasing the system size N the ratio B/N would go to zero. In either case the reported value does not bring much information.

After the authors address this issue, the manuscript can be reconsidered.

Reviewer #2 (Remarks to the Author):

The authors present a thorough and detailed theoretical study of the phase transition mechanism between beta and lambda Ti₃O₅, a material with high technological impact.

By a combination of direct molecular dynamics and meta dynamics simulations based on a carefully trained machine-learned potential they show that the phase transition is not a concerted process but localized along a certain lattice plane. Their simulations qualitatively reproduce experimentally observed trends with temperature and pressure.

While the study is theoretically and technically of high quality and the results give relevant physical insight, there are several aspects that have to be discussed before the manuscript can be accepted for publication in Nature Materials.

General comments:

1. The authors discount a very recent publication <https://doi.org/10.1021/acs.jpcc.3c04986>), where the general phase transition mechanism was also theoretically studied, with similar methods.

The novel aspects of the present study should be discussed with respect to this earlier work.

2. The comparison with the previous study should also contain an explanation for the choice of the standard functional PBE. It was shown that the meta-GGA functional r2SCAN provides improved solid-state properties of both Ti₃O₅ phases. A thorough comparison of calculated and measured thermodynamical data of the two phases should be provided.

3. Furthermore it should be discussed why long-range dispersion was not taken into account. London dispersion plays a significant role in calculations of the relative phase stability.

4. From experiment it is known that both Ti₃O₅ phases have magnetic ground states. The present study does not mention how this was treated. This needs a detailed explanation.

Specific comments:

1. Methods section: The authors claim that the fitted potential also well describes alpha-Ti₃O₅, because of its structural similarity with lambda-Ti₃O₅. However, in Figure S1c the phonon dispersion for alpha-Ti₃O₅ is not accurate for the soft (imaginary) mode and less precise in general. This should be discussed.

2. Methods section: Figure S2: Surprisingly, the validation set exhibits a significantly lower error compared to the training set. This should be explained.

3. The phonon dispersion plots do not provide significant information due to the high density of bands. The number of bands should be reduced in the presentations.

4. It would be interesting to see the B_u "transition phonon modes" as a function of temperature and/or pressure, to get more insight into the phase transition.

5. The authors should highlight their rather accurate predicted phase transition temperature obtained by WMetaD (Figure S4b) with the MLP in the main text.

In general the computed data should be compared to experimental data.

Summary of changes

- (1) We have updated Figure 1 to include the structure factor, time evolution of the collective variable, and temperature-dependent free energy difference so that the predicted accurate phase transition temperature is highlighted. Accordingly, the Supplementary Figure 4 has been updated and the main text has been modified to reflect these changes.
- (2) We have performed new phonon calculations for α -Ti₃O₅ at 600 K to demonstrate that our developed machine learning potential is indeed suited for accurately describing the anharmonic phonon-phonon interactions. This results in the updated Supplementary Figure 1. We have also updated the Methods section to explain the methodology used for anharmonic phonon calculations and referenced the corresponding methodology papers (i.e., Ref. [66] for the temperature-dependent effective potential method and Ref. [67] for the ALAMODE code).
- (3) We have expanded the validation dataset, performed the kernel principal component analysis map of training structures and validation structures, and updated the validation errors. This leads to the updated Supplementary Figure 2.
- (4) We have carefully carried out phonon mode analysis and clarified the decisive phonon mode that drives the phase transformation from β - to λ -Ti₃O₅, and vice versa. Moreover, we have examined the pressure- and temperature-dependent phonon modes. This leads to the updated Supplementary Figure 3. The relevant discussions have been added in the revised main text.
- (5) We have performed new variable-cell climbing image nudged elastic band calculations for increasing system sizes and clarified the system size dependence of energy barrier for collective and nucleation mechanisms. The results have been reflected in the revised main text and the new Supplementary Figure 18.
- (6) We have updated Supplementary Table 1 with new calculated results and added a new Supplementary Note to rationalize the choice of density functional and the treatment of magnetism.
- (7) We have updated the phonon dispersions plots with reduced linewidth for a better presentation and provided the raw data in the Source Data (in excel format).
- (8) We have added relevant discussions on the very recent publications of Jütten and Bredow (Refs. [27] and [28]) in the main text (i.e., the “Introduction” part, the Section C of “Results” part, and “Discussion” part).
- (9) We have made minor adjustments to the “Abstract”, “Data Availability”, and “Code Availability” Sections to adhere to the requirements set by the journal.

All major changes made in the main text and Supplementary Information have been highlighted in blue color.

Response to Reviewer A

Reviewer: The work is interesting, original and timely. The work appears technically correct and the results are likely reliable and interesting for the solid-state physics and materials science community. However, I see a problem with the interpretation of the results. The authors say on p. 13 "By contrast, the transformation via the in-plane nucleation is energetically more favorable, with a decreased energy barrier of 0.07 eV/f.u. (see Path II in Fig. 5b)." I have the following comments to this statement.

Authors: We are grateful to the reviewer for taking the time and effort to read our manuscript. We also thank the reviewer very much for his/her positive remarks. Below we address in detail all the points raised.

Reviewer: In order to claim that a real nucleation event is observed, one has to show that the barrier B is independent on the system size, which was not the case here. It would be necessary to perform several calculations for larger system sizes. Dividing the barrier B by the number of formula units N and reporting energy per formula unit B/N makes sense in case of collective mechanism when B/N stays constant for increasing N , not in the case of nucleation. If the barrier indeed scales linearly with the system size and B/N stays constant, then what is observed is not nucleation. If instead a real nucleation is observed, with a finite barrier B , by increasing the system size N the ratio B/N would go to zero. In either case the reported value does not bring much information.

Authors: We thank the reviewer for his/her insightful comments. We fully agree with the reviewer. Following the suggestions of the reviewer, we carried out several new variable-cell climbing image nudged elastic band calculations (CI-NEB) with increasing system sizes. The results are shown in Fig. R1. It can be seen that for the case of a collective mechanism, the energy barrier B increases linearly with the system size N , leading to a constant ratio B/N with respect to N , as the reviewer pointed out. However, for the case of the nucleation mechanism, the energy barrier B is almost independent of the system size N , resulting in a zero B/N ratio limit when the system size N is large. We would like to note that the convergence of variable-cell CI-NEB calculations for solid-solid transformations is known to become very difficult when the system size is large with many (lattice and atomic positions) degrees of freedom. Therefore, we restrict our calculations to the system with a maximum of 768 atoms. All these new calculations clearly indicate that the transformation from β - to λ -Ti₃O₅ indeed follows a nucleation mechanism, which is kinetically more favored than the collective mechanism.

In the revised version of our manuscript, we have added the relevant discussions on this point, which have been annotated in blue color. For a clearer quantitative comparison of the energy barriers of the two transformation mechanisms, we keep the use of reduced energy (i.e., energy per formula unit) for the employed specific system size in Fig. 5 of the main text, but remind the readers that for the case of the nucleation mechanism the energy barrier is independent of the system size in the main text. In addition, we have added Fig. R1 to the Supplementary Fig. 18.

Fig. R1. **System size dependence of energy barrier.** **a**, Calculated energy barrier as a function of system size for two different transformation mechanisms from β to λ phases, i.e., collective mechanism (Path I) and nucleation mechanism (Path II). Note that for a better presentation the results for the case of the collective mechanism beyond the system size of 500 atoms are not shown. **b**, Calculated ratio between the energy barrier and the system size as a function of system size.

Reviewer: After the authors address this issue, the manuscript can be reconsidered.

Authors: We thank the reviewer again for the positive remarks and insightful comments. With these appropriate modifications, we hope that our manuscript is now suitable for publication.

Response to Reviewer B

Reviewer: The authors present a thorough and detailed theoretical study of the phase transition mechanism between beta and lambda Ti_3O_5 , a material with high technological impact.

By a combination of direct molecular dynamics and meta dynamics simulations based on a carefully trained machine-learned potential they show that the phase transition is not a concerted process but localized along a certain lattice plane. Their simulations qualitatively reproduce experimentally observed trends with temperature and pressure.

While the study is theoretically and technically of high quality and the results give relevant physical insight, there are several aspects that have to be discussed before the manuscript can be accepted for publication in Nature Communications.

Authors: We thank the reviewer very much for his/her careful and expert reading of our manuscript, for recognizing the high quality of our study, and for supporting the publication of our work as long as we suitably address the aspects raised by the reviewer. We also thank the reviewer for the useful comments and suggestions. These certainly helped us to improve the presentation of our manuscript. Below we address all the comments and questions.

Reviewer: The authors discount a very recent publication <https://doi.org/10.1021/acs.jpcc.3c04986>), where the general phase transition mechanism was also theoretically studied, with similar methods. The novel aspects of the present study should be discussed with respect to this earlier work.

Authors: We sincerely express our gratitude to the reviewer for bringing a new publication to our attention. The paper was released very recently, and we apologize for inadvertently overlooking it in the initial version of our manuscript. It is right that the work of Jütten and Bredow [J. Phys. Chem. C, 127, 20530 (2023)] also theoretically investigated the phase transition between β - and λ - Ti_3O_5 phases using $r^2\text{SCAN-D3}$ based machine learning potential (MLP) and provided important insights on the pressure effect on the phase transition. As compared to this work, the novelties of our study lie in the following aspects:

(i) The work of Jütten and Bredow focused on the pressure effect on the phase transition. In the present work, we not only consider the pressure effect, but also study the temperature effect. Moreover, using a designed collective variable for the metadynamics simulations, **we have for the first time built a full *ab initio* thermodynamic pressure-temperature phase diagram, in good agreement with experiment.** This is enabled by developing an efficient MLP with near first-principles accuracy through a combination of an on-the-fly active learning method and the state-of-the-art advanced sampling technique.

(ii) The MLP generated in the work of Jütten and Bredow was mainly used to optimize the interface models with mixed phases. In the present work, we aimed to explore the underlying microscopic transition mechanism, particularly the kinetic pathway for the ultrafast and reversible transition. **Through metadynamics simulations and large-scale molecular dynamics simulations, we have discovered a novel hitherto-unreported kinetically favorable in-plane nucleated multistep mechanism for the β - Ti_3O_5 to λ - Ti_3O_5 phase transition.** This multistep barrier-lowering kinetic process has not been revealed before and naturally elucidates the underlying reason why an ultrafast and reversible phenomenon can manifest in a reconstructive solid-solid phase transition.

(iii) **Our study has yielded an interesting finding that all superlattices consisting of any combination of β -like and λ -like structural motifs along the c axis are dynamically stable.** To the best of our knowledge, we are not aware of the presence of robust dynamical stability in a periodic solid formed by any combination of two different structural motifs. This unique flexibility opens avenues for tuning functional properties, e.g., insulator-to-metal transition and specific heat modulation. Consequently, our findings hold great potential for practical applications in the design of novel energy storage media, optical storage media and sensor devices.

In the revised version of manuscript, we have added relevant discussion on the very recent publication of Jütten and Bredow [Ref. [27], J. Phys. Chem. C, 127, 20530 (2023)] (i.e., the “Introduction” part, the Section C of “Results” part, and “Discussion” part).

Reviewer: The comparison with the previous study should also contain an explanation for the choice of the standard functional PBE. It was shown that the meta-GGA functional r^2 SCAN provides improved solid-state properties of both Ti_3O_5 phases. A thorough comparison of calculated and measured thermodynamical data of the two phases should be provided.

Authors: We thank the reviewer for the useful comment. In the original Supplementary Table 1, we provided the predicted lattice parameters and energy differences at 0 K for β , λ and α phases. As compared to the PBEsol and the SCAN functionals, the standard PBE functional yields an excellent description of the lattice parameters of all three phases. PBE is, however, much cheaper and only needs a fraction of the computational of the meta-GGA SCAN functional. The meta-GGA functional r^2 SCAN is a variant of SCAN functional, which improves the numerical stability over SCAN but at the cost of slightly deteriorating the accuracy. According to the work of Jütten and Bredow [J. Phys. Chem. C, 126, 7809 (2022)], r^2 SCAN+D3 indeed yields good description of structural and thermodynamical properties of β and λ phases. In Table R1 below, we compare the calculated and measured phase transition temperature T_c and phase transition enthalpy ΔH . It can be seen that PBE using a nonmagnetic setup and with the anharmonic phonon-phonon interactions fully accounted for yields a T_c of 535 K and ΔH of 0.091 eV/f.u., in good agreement with the predictions by the r^2 SCAN+D3 method using the much simpler harmonic approximation ($T_c=525$ K and $\Delta H=0.098$ eV/f.u.) [Jütten and Bredow, J. Phys. Chem. C, 126, 7809 (2022)]. Both are in line with the experimental results ($T_c=470$ K and $\Delta H=0.124 \pm 0.01$ eV/f.u.) [Tokoro *et al.*, Nature Communications 6, 7037 (2015)] (see Table R1). We emphasize that fully considering the anharmonic phonon-phonon interactions is necessary to obtain accurate phase transition temperature and enthalpy (see Table R1). This has also been highlighted in the work of Trail *et al.*, Phys. Rev. B 95, 121108 (2017) for improved descriptions of the thermodynamic properties of titanium suboxides. We note that the values reported by Jütten and Bredow using r^2 SCAN+D3 adopted the simple harmonic approximation. Including anharmonic contributions would lower the temperature likely to around 300 K. In addition, we found that for the case of the collective phase transformation from β to λ phases, the standard PBE functional with nonmagnetic setups yields a comparable energy barrier (0.25 eV/f.u.) as r^2 SCAN+D3 [0.27 eV/f.u., J. Phys. Chem. C, 126, 7809 (2022)]. All these comparisons indicate that the standard PBE functional using a nonmagnetic setup is a very reasonable choice for describing the structural and thermodynamical properties of β and λ phases, likely even superior to r^2 SCAN+D3.

We want to finish this paragraph with a general note: as it stands, the present density functionals

require a sizable degree of unavoidable empiricism. Whenever materials with strong correlation effects need to be described using a mean field method such as DFT, it is *a priori* not clear what is the best mean field description. Antiferromagnetic ordering is at best a surrogate for a paramagnetic material. In fact, dynamical mean field theory (DMFT) calculations are always initialized from the nonmagnetic ground state, and not from the antiferromagnetically ordered state, for good reasons: if DMFT predicts an antiferromagnetic ground state, then this corresponds to a complicated multi-determinantal ground state that leads, on average, to a *nonmagnetic solution* on all metal atoms, similar to a singlet.

In the revised version of our manuscript, we have added some of the relevant discussions and added comparison with the results predicted by r²SCAN+D3 as well as the experimental values in the revised Supplementary Table 1. Furthermore, we have added a Supplementary Note to rationalize the choice of the density functional.

Table R1. Calculated and measured phase transition temperature and phase transition enthalpy. PBE-NM denotes the PBE functional using a nonmagnetic setup.

	Phase transition temperature T_c (K) at 0 GPa	Phase transition enthalpy ΔH (eV/f.u.)
PBE-NM (Harmonic approximation)	728	0.080
PBE-NM (Quasi-harmonic approximation)	822	0.084
PBE-NM (Fully anharmonic, MD)	535	0.091
r ² SCAN+D3 (Harmonic approximation) [Jütten and Bredow, J. Phys. Chem. C, 126, 7809 (2022)]	525	0.098
Experiment [Tokoro et al. , Nature Communications 6, 7037 (2015)]	470	0.124±0.01

Reviewer: Furthermore, it should be discussed why long-range dispersion was not taken into account. London dispersion plays a significant role in calculations of the relative phase stability.

Authors: We thank the reviewer for the comment. Generally, the D3 dispersion correction method is not accurate for ionic materials: in Ti₃O₅ charge is transferred from the Ti atoms to the oxygen atoms, but D3 and many other *posterior* corrections have not been designed to take this charge transfer into account (the minimum complexity to describe this is the Tkatchenko-Scheffler method with iterative Hirshfeld partitioning). Furthermore, the D3 corrections have been parameterized to describe interactions through the vacuum, whereas in bulk materials the dispersion corrections are strongly screened by the surrounding atoms. The minimum level of complexity are thus many-body-dispersion corrections (MBD) that account for the interplay between the polarizable entities. There is no reason to believe that D3 captures the relevant physics in Ti₃O₅. Hence, adding them on top of a magnetic calculation adds at least the same level of empiricism as remaining from the outset with simpler functionals such as PBE. Occam's Razor in fact advocates adopting the simplest possible

solution. What the present work clearly demonstrates is that we can obtain a very reasonable phase transition temperature from the simplest plausible mean field calculation. This is very remarkable, and worthwhile for a much wider screening of phase transitions in materials that are considered to be strongly correlated!

According to the work of Jütten and Bredow [J. Phys. Chem. C, 126, 7809 (2022)], the majority of assessed functionals yield qualitatively wrong negative values for the energy difference at 0 K between the λ -Ti₃O₅ and β -Ti₃O₅ phases without dispersion corrections (i.e., they predict the λ -phase to be more stable than the β -phase). It was found that a correct positive energy difference is only restored when dispersion corrections are considered. As discussed above, we speculate that the reason causing such behavior is likely due to the inclusion of the magnetic degrees of freedom in a static mean-field calculation, which often leads to an incorrect electronic ground state. For instance, most of the assessed functionals (except for B97H-D3 and r²SCAN-D3) predict the β -Ti₃O₅ phase to be an antiferromagnetic (AFM) semiconductor or insulator, but predict the λ -Ti₃O₅ phase to be a ferromagnetic semiconductor [see Supplementary Table S19 of J. Phys. Chem. C, 126, 7809 (2022)]. However, these are not consistent with the experimental results that the β phase is a nonmagnetic semiconductor, while the λ phase is a weak Pauli paramagnetic metal (see also the response to the next comment of the reviewer). To check this point, we calculated the energy difference between the two phases using the PBE0 and HSE06 hybrid functionals adopting the nonmagnetic setups. The results are compiled in Table R2. It can be seen that using the nonmagnetic setups, even PBE0 or HSE06 without dispersion corrections are able to yield correct positive energy difference between the two phases. Including the dispersion corrections almost doubles the energy difference between the two phases, worsening the agreement with the experimental value.

Table R2. Calculated energy difference at 0 K $\Delta E(\lambda-\beta)$ (in eV/f.u.) between λ - and β -Ti₃O₅ by different methods. For reference, we note that the experimentally measured phase transition enthalpy is 0.124 ± 0.01 eV/f.u. [Tokoro *et al.*, Nature Communications 6, 7037 (2015)].

$\Delta E(\lambda-\beta)$ (eV/f.u.)	PBE0	PBE0+D3	HSE06	HSE06+D3
This work (Nonmagnetic setup)	0.155	0.302	0.158	0.305
Jütten and Bredow, J. Phys. Chem. C, 126, 7809 (2022) (Considering magnetism)	-0.169	0.091	-0.178	0.081

We also examined the effect of dispersion corrections on the structural and thermodynamical properties of both phases at the PBE level. We found that although both PBE and PBE+D3 give a positive energy difference between the two phases, PBE+D3 yields a significantly large value of 0.212 eV/f.u., far larger than the experimentally measured phase transition enthalpy [0.124 ± 0.01 eV/f.u., Tokoro *et al.*, Nature Communications 6, 7037 (2015)] (see the updated Supplementary Table 1). In addition, we found that inclusion of the D3 correction slightly deteriorates the description of lattice parameters (see Supplementary Table 1). By contrast, the standard PBE functional yields excellent structural and thermodynamical properties of both phases. The predicted thermodynamical temperature-pressure phase diagram using the PBE-derived MLP is in good agreement with experiment (see Fig. 1 of the main text).

Finally, and maybe most importantly, we would like to emphasize that although the choice of the

density functional would to some extent quantitatively change the actual values of the phase transition temperature and pressures, obtaining an accurate thermodynamical temperature-pressure phase diagram is just one part of the present study. Our main discovery of unusual layer-by-layer phase transformation initiated by kinetically favorable in-plane nucleated mechanism will remain unchanged by the change of the density functional. We have added relevant discussion on this point in the Supplementary Note.

Reviewer: From experiment it is known that both Ti_3O_5 phases have magnetic ground states. The present study does not mention how this was treated. This needs a detailed explanation.

Authors: We thank the reviewer for the comment. Fig. R2a shows the magnetic susceptibility (χ) versus temperature graph of the flake form λ - Ti_3O_5 and single-crystal β - Ti_3O_5 under an external field of 0.5 T [Ohkoshi *et al.*, Nature Chemistry 2, 539 (2010)]. λ - Ti_3O_5 exhibit χ values around 2×10^{-4} e.m.u. per Ti atom, indicating that λ - Ti_3O_5 is a weak Pauli paramagnet because of metallic conduction. According to Ohkoshi *et al.*, the gradual decrease below 150 K is due to the spin-orbit coupling on the Ti^{3+} ion, and the rapid increase below 30 K is attributed to a small amount of Curie paramagnetism (about 0.1%), likely caused by defects in the material. By contrast, β - Ti_3O_5 exhibits nearly vanishing χ values, indicating its nonmagnetic behavior. Fig. R2b shows the experimental valence band spectra in the band gap region beneath the Fermi level for the β (blue dots) and λ (red dots) phases, respectively [Kobayashi *et al.*, Phys. Rev. B 95, 085133 (2017)]. One can see that the onset of the λ -phase spectrum just coincides with the Fermi level, whereas the onset of the β -phase spectrum is 0.2 eV below the Fermi level. This indicates that the λ -phase is a metal, whereas the β -phase is a semiconductor with a gap of 0.2 eV.

Fig. R2. **a**, Magnetic susceptibility versus temperature graph of the flake form λ - Ti_3O_5 (red line) and single-crystal β - Ti_3O_5 (dashed black line) under an external field of 0.5 T [Taken from Ohkoshi *et al.*, Nature Chemistry 2, 539 (2010)]. **b**, Experimental valence band spectra in the band gap region beneath the Fermi level for the β (blue dots) and λ (red dots) phases, respectively [Taken from Kobayashi *et al.*, Phys. Rev. B 95, 085133 (2017), Copyright 2017 American Physical Society].

Based on these experimental results, we therefore neglected the magnetic degrees of freedom and adopted nonmagnetic calculations throughout the work. This is a reasonable choice, because

(i) The density functional theory calculations employing nonmagnetic setups yield good descriptions of electronic structures of both phases, e.g., the formation of a bipolaron (with no spin) of Ti_3 - Ti_3 caused by σ -type bonding of d_{xy} orbitals of Ti_3 atoms in β - Ti_3O_5 and the formation

of slipped π -stacking between the d_{xy} orbital on Ti2 and the d_{xy} orbital on Ti3 in λ -Ti₃O₅ [Ohkoshi *et al.*, Nature Chemistry 2, 539 (2010), Kobayashi *et al.*, Phys. Rev. B 95, 085133 (2017), Yang *et al.*, Nature, 622, 499 (2023)].

(ii) The density functionals considering the magnetic setups often predict incorrect ground states that are not consistent with experiments. For instance, according to the work of Jütten and Bredow [J. Phys. Chem. C, 126, 7809 (2022)], hybrid functional M06-D3 predicts the β -Ti₃O₅ phase to be an antiferromagnetic (AFM) semiconductor, while predicts the λ -Ti₃O₅ phase to be a ferromagnetic semiconductor. The r^2 SCAN-D3 method yields ferromagnetic metals for both β - and λ -Ti₃O₅ phases [J. Phys. Chem. C, 126, 7809 (2022)], while the PBE+ U method predicts that the AFM order was found to be the ground states for both phases [Marré *et al.*, Nature Communications 12, 1239 (2021)]. With $U=4.4$ eV the PBE+ U method predicts an AFM insulator for all the three phases (β , λ , and α) and even yield qualitatively wrong negative values for the energy difference between the two phases (see the updated Supplementary Table 1). We kindly recall that in experiment the β phase is a nonmagnetic semiconductor, while the λ phase is a weak Pauli paramagnetic metal (Fig. R2).

(iii) The PBE functional with the nonmagnetic setups gives an excellent description of the experimental lattice parameters of β , λ as well as α phases as compared to other density functionals such as PBEsol and SCAN (see Supplementary Table 1).

(iv) Considering the magnetic degrees of freedom hardly changes the thermodynamical properties of both phases [Jütten and Bredow, J. Phys. Chem. C, 126, 7809 (2022)]. On the one hand, the calculated energy differences between different spin configurations are found to be small [see Table 1 of J. Phys. Chem. C, 126, 7809 (2022)]. On the other hand, the activation barriers from β to λ only change slightly between different magnetic configurations [see Table 3 of J. Phys. Chem. C, 126, 7809 (2022)]. We also compared the phase transition temperature T_c and phase transition enthalpy ΔH predicted by our PBE-derived MLP to the results predicted by r^2 SCAN+D3 as well as the experimental data. It can be seen from Table R1 that the MLP fully accounting for the anharmonic phonon-phonon interactions yields a T_c of 535 K and ΔH of 0.091 eV/f.u., in good agreement with the predictions by the r^2 SCAN+D3 method within the simpler harmonic approximation ($T_c=525$ K and $\Delta H=0.098$ eV/f.u.) [Jütten and Bredow, J. Phys. Chem. C, 126, 7809 (2022)]. Both are in line with the experimental results ($T_c=470$ K and $\Delta H=0.124\pm 0.01$ eV/f.u.) [Tokoro *et al.*, Nature Communications 6, 7037 (2015)]. However, PBE is computationally cheaper and needs a fraction of the computational cost of the meta-GGA r^2 SCAN functional. Furthermore, including the magnetic degree of freedom will dramatically increase the complexity of MLP model. As far as we know, only few works currently attempt at incorporating the spin descriptor besides the structural descriptor into the MLP model [Eckhoff *et al.*, npj Computational Materials 7, 170 (2021); Novikov *et al.*, npj Computational Materials 8, 13 (2022); Chapman *et al.*, Scientific Reports 12, 22451 (2022); Yu *et al.*, arXiv:2203.02853 (2023)].

In our revised manuscript, we have added this information associated with the treatment of magnetic degree of freedom in the Methods Section (annotated in blue color) and also added a Supplementary Note for more detailed discussions.

Reviewer: Specific comments: Methods section: The authors claim that the fitted potential also well

describes α -Ti₃O₅, because of its structural similarity with λ -Ti₃O₅. However, in Figure S1c the phonon dispersion for α -Ti₃O₅ is not accurate for the soft (imaginary) mode and less precise in general. This should be discussed.

Authors: We thank the reviewer for the comment. In the present work, we aimed to describe the unusual layer-by-layer phase transformation between the β - and λ -Ti₃O₅ phases, and therefore, the training datasets mostly cover the relevant phase space of the β - and λ -Ti₃O₅ phases (not the high-temperature α -Ti₃O₅ phase). In other words, the prediction of phonon dispersion relations of the α -Ti₃O₅ phase by the developed MLP is somewhat a by-product. However, we argued that due to the structural similarity between α -Ti₃O₅ and λ -Ti₃O₅, the MLP predicted phonon dispersion relations exhibit good agreement with the DFT calculations. Although a quantitative discrepancy exists, the overall phonon modes are already reproduced by the MLP, in particular the presence of imaginary phonon modes at 0 K.

We would also like to note that in general, harmonic vibrational properties are already a stringent test of the quality of MLPs, especially when considering materials with soft phonon modes. Notably, George *et al.* [J. Chem. Phys. 153, 044104 (2020)] highlights the difficulties of training ML potentials that accurately describe phonon modes. As a solution, a rather involved procedure for building a reference database was proposed. While the description of phonon dispersion relations requires only a two-body term (the harmonic force constants), anharmonicity, on the other hand, requires the development of a surrogate model for tiny, subtle energy differences on the highly corrugated harmonic energy surface. The MLP must capture at least three-body anharmonic force constants, the coupling between phonons and lattice distortions, and, especially at higher temperature, even four-particle interactions. In order to demonstrate that our developed MLP is indeed suited for accurately describing the anharmonic phonon-phonon interactions, we computed the phonon dispersion relations of α -Ti₃O₅ at a high temperature of 600 K (above the phase transition temperature). The results along with the harmonic phonon dispersion relations predicted at 0 K are shown in Fig. R3. It is evident that the imaginary phonon modes present at 0 K disappear at the high temperature. This is expected, since the α -Ti₃O₅ is a high-temperature phase and becomes dynamically stable only at high temperature. This test clearly indicates that the developed MLP is accurate and is capable of describing the anharmonic phonon-phonon interactions of α -Ti₃O₅.

In the revised version of our manuscript, we have added the relevant discussions and computational details in the main text (annotated in blue color) and included the Fig. R3 as part of the revised Supplementary Fig. 1.

Fig. R3. Phonon dispersion relations of α -Ti₃O₅ predicted by **a**, MLP and DFT at 0 K, and **b**, MLP at 600 K. The anharmonic phonon dispersions at 600 K were obtained using temperature dependent effective potential method [Hellman *et al.*, Physical Review B, 84(18), 180301 (2011)] and a supercell of 640 atoms. Specifically, a 1 ns equilibrium MD run was firstly conducted in an isothermal-isobaric ensemble to obtain the thermal equilibrium structure at 600 K. Subsequently, a 500 ps MD run in a canonical ensemble was carried out using the equilibrated structure. During the MD run, displacements and forces were sampled in regular intervals of 0.5 ps. Finally, the anharmonic force constants were extracted using the ALAMODE code [Tadano *et al.*, J. Phys.: Condens. Matter 26, 225402 (2014)].

Reviewer: Methods section: Figure S2: Surprisingly, the validation set exhibits a significantly lower error compared to the training set. This should be explained.

Authors: We thank the reviewer for the useful comment. This indeed needs an explanation. As mentioned in the Methods Section of our manuscript, the developed MLP potential was validated on a test dataset containing 378 structures of 96 atoms, which were randomly chosen from a metadynamics trajectory at 600 K using the generated MTP potential. As compared to the training dataset which contains 3 775 structures of 96 atoms and covers a wide range of the phase space, the phase space represented by the validation dataset is clearly limited, as demonstrated by the kernel principal component analysis in Fig. R4a. This is the reason why the validation errors are relatively small as compared to the training errors.

In the revised version of our manuscript, we expanded the validation dataset up to 748 structures of 96 atoms. Despite being still not as representative as the training dataset, the updated validation dataset now covers more phase space, as illustrated by Fig. R4b, where more structures along the transition path are sampled. Due to the expansion of the validation dataset, the corresponding validations errors are slightly increased towards the training errors (see Table R3). We have updated the Supplementary Fig. 2 with the newly expanded validation dataset and also included the kernel principal component analysis for the training and validation datasets.

Fig. R4. The kernel principal component analysis map of training structures (blue squares) and validation structures (red squares). **a**, Original validation dataset. **b**, Expanded validation dataset.

Table R3. Training and validation errors for energies, forces and stress tensors.

Root-mean-square-errors (RMSEs)	Training dataset (3775 structures of 96 atoms)	Validation dataset (Original, 378 structures of 96 atoms)	Validation dataset (Expanded, 748 structures of 96 atoms)
Energy (meV/atom)	2.01	1.64	1.64
Force (eV/Å)	0.119	0.108	0.124
Stress tensor (GPa)	0.151	0.150	0.192

For the convenience of the reviewer, the revised Supplementary Fig. 2 is also given below.

Figure 2. **Training and validation datasets and MLP predictions vs. DFT results.** **a**, The kernel principal component analysis map of training structures (blue squares) and validation structures (red squares). **b**, MLP predicted energies vs. DFT results. **c**, MLP predicted forces vs. DFT results. **d**, MLP predicted stress tensors vs. DFT results.

Reviewer: The phonon dispersion plots do not provide significant information due to the high density of bands. The number of bands should be reduced in the presentations.

Authors: We thank the reviewer for the comment. Due to the large number of atoms in the unit cell in particular for those metastable phases consisting of combination of β - and λ -like local structural

motifs, the number of phonon bands is indeed high, leading to the overcrowding in the presentation for limited space. To make the phonon bands clearer, we have now reduced the linewidth for all the phonon dispersions plots (e.g., Supplementary Fig. 8, Supplementary Fig. 15 and Supplementary Fig. 17). We would like to note that although the details of phonon bands are important by themselves, the presentation of phonon bands aims to highlight the absence of imaginary phonon modes for the metastable phases. Therefore, we prefer to provide the full phonon bands in the plots. However, we have provided the raw phonon bands data as the Source Data (in excel format) for interested readers, which have been stated in the “Data Availability” Section.

Reviewer: It would be interesting to see the B_u "transition phonon modes" as a function of temperature and/or pressure, to get more insight into the phase transition.

Authors: We thank the reviewer for the insightful comment. This reminds us to take a closer inspection of phonon modes. We realized that the B_u mode at Γ previously identified in our original manuscript and also in the paper of Tokoro *et al.* [Nature Communications 6, 7037 (2015)] is not capable of deriving the phase transformation from β to λ , and vice versa. Specifically, the B_u mode does not lead to the rotation of Ti3-Ti3 dimers, which is the precursor for the phase transformation. Instead, the actual phonon mode that can drive the phase transformation is the A_g phonon mode. In Fig. R5a we show the detailed transformation process by following the eigenvector of the A_g phonon mode.

Following the suggestions of the reviewer, we also carried out pressure- and temperature-dependent phonon dispersions calculations. The results along with the pressure- and temperature-dependent A_g mode frequencies and system volumes are compiled in Fig. R5. It can be seen from Fig. R5f that the A_g phonon mode frequency of λ -Ti₃O₅ first increases up to 3 GPa mostly due to the reduction of the system volume. As the pressure increases, the A_g phonon mode frequency decreases. That is, the phonon softening associated with the phase transition dominates the phonon hardening due to the volume contraction. This is consistent with the experimental observation that increasing pressure is beneficial to the $\lambda \rightarrow \beta$ phase transition. Turning to the temperature-dependent phonon dispersions of the β phase, we can see that the A_g phonon mode frequency becomes soft as the temperature increases (Fig. R5k), in line with the experimental result that increasing temperature is helpful for the $\beta \rightarrow \lambda$ phase transition.

We have added relevant discussions in the revised main text (annotated in blue color) and replaced the original Supplementary Fig. 3 with Fig. R5.

Fig. R5. **Phonon mode analysis.** **a**, The decisive phonon modes that drive the phase transformation from β to λ , and vice versa. The arrows on the atoms indicate the atomic displacements associated with the specified phonon mode. **b-e**, Pressure-dependent phonon dispersions of λ - Ti_3O_5 . **f**, Pressure-dependent A_g phonon mode frequencies and system volumes of λ - Ti_3O_5 . **g-j**, Temperature-dependent phonon dispersions of β - Ti_3O_5 . **k**, Temperature-dependent A_g phonon mode frequencies and system volumes of β - Ti_3O_5 . Note that the red dots in the plots of phonon dispersions indicate the A_g phonon mode.

Reviewer: The authors should highlight their rather accurate predicted phase transition temperature obtained by WTMetaD (Figure S4b) with the MLP in the main text. In general, the computed data should be compared to experimental data.

Authors: We thank the reviewer for the useful comment. Following the reviewer's suggestion, we have moved parts of Supplementary Fig. 4 (e.g., structure factor and prediction of transition temperature) to the Fig. 1 of the main text and modified the main text accordingly to highlight the good agreement between the calculated phase transition temperature and phase transition enthalpy and the experimental data. For the convenience of the reviewer, the revised Fig. 1 is also given below.

Figure 1. **Metadynamics simulations and phase diagram.** **a**, Simulated XRDs of β - and λ - Ti_3O_5 using a 96-atom supercell. **b**, Time evolution of collective variable at different temperatures and 0 GPa. **c**, Free energy difference $\Delta G_{\lambda-\beta}$ calculated by the well-tempered variant of metadynamics (WTMetaD) and the quasi-harmonic approximation (QHA) as a function of temperature. The error bar indicates the standard deviation from three independent simulations. **d**, Pressure-temperature phase diagram predicted by metadynamics simulations. Insets show the crystal structures of β - and λ - Ti_3O_5 , where the black lines indicate the unit cell.

Reviewer #2 (Remarks to the Author):

The authors have addressed the issues raised in the first review and I recommend the manuscript for publication in Nature Communications.

Reviewer #3 (Remarks to the Author):

The authors have replied to all of my previous comments. They provide acceptable arguments and complementary data to justify the choice of their methodology, and extended their manuscript accordingly. Furthermore they discuss recent literature and detail the new physical insights provided by their work.

For these reasons I recommend the revised manuscript for publication.

Response to Reviewer #2

Reviewer: The authors have addressed the issues raised in the first review and I recommend the manuscript for publication in Nature Communications.

Authors: We thank the reviewer for taking the time to review our manuscript and recommending it for publication.

Response to Reviewer #3

Reviewer: The authors have replied to all of my previous comments. They provide acceptable arguments and complementary data to justify the choice of their methodology, and extended their manuscript accordingly. Furthermore they discuss recent literature and detail the new physical insights provided by their work. For these reasons I recommend the revised manuscript for publication.

Authors: We thank the reviewer for taking the time to review our manuscript and recommending it for publication.